

# Prediction of microbe-drug associations using a CNN-Bernoulli random forest model

Zihao Song[1], Qingnuo Li[1], Jincheng Zhao[1], Qinggang Bu[1], Zekang Bian[2] and Jia Qu[1]

[1] The School of Computer Science and Artificial Intelligence & Aliyun School of Big Data, Changzhou University, Changzhou, China
[2] The School of AI & Computer Science, Jiangnan University, Wuxi, China

Corresponding author
Jia Qu, tb17060015b4@cumt.edu.cn

## ABSTRACT

**Background**. Antibiotics play a critical role in treating microbial infections. However, their widespread use has contributed to the growing problem of microbial resistance. Addressing this challenge requires the identification of new microbe-drug associations to support the development of novel therapeutic strategies. Since traditional wet-lab experiments are time-consuming and costly, computational models offer an efficient alternative for discovering potential applications of existing drugs against previously untested microbes. These models can facilitate the identification of novel microbe-drug associations and help counteract resistance mechanisms.

**Methods**. This study proposes a novel computational model: convolutional neural network with Bernoulli random forest for Microbe-Drug Association prediction (CNNBRFMDA). The model constructs feature vectors for all microbe-drug pairs based on known associations, microbe similarity, and drug similarity. A subset of these vectors is randomly selected to form the training set. A convolutional neural network (CNN) is then used to reduce the dimensionality of all feature vectors, including those in the training set. The reduced training set is subsequently used to train a Bernoulli random forest (BRF) to predict potential microbe-drug associations. The innovation of CNNBRFMDA lies in its integration of CNN for nonlinear feature extraction and BRF for robust prediction. This approach enhances computational efficiency and improves the model's ability to capture complex patterns, thereby increasing the precision and interpretability of drug response predictions. The dual use of the Bernoulli distribution in BRF ensures algorithmic consistency and contributes to superior performance.

**Results**. The model was evaluated using five-fold cross-validation on the Microbe-Drug Association Database (MDAD) and abiofilm datasets. CNNBRFMDA achieved mean AUC scores of 0.9017 ± 0.0032 (MDAD) and 0.9146 ± 0.0041 (abiofilm). Two case studies further validated the model's reliability: 41 of the top 50 predicted microbes associated with ciprofloxacin and 38 of the top 50 associated with moxifloxacin were confirmed through literature review.

# INTRODUCTION

Microbes are a class of tiny organisms that include bacteria, viruses, fungi, and parasites (*Schommer & Gallo, 2013*). They are widely distributed in our surrounding environment and played important roles in the lives of humans and animals (*Rashno et al., 2021*). Some of microbes are beneficial to human bodies and ecosystems, some of microbes are pathogens that can cause serious diseases (*Tremlett & Waubant, 2018*).

On the one hand, some microbes have evolved with humans and work in concert with the human body to maintain human immune and metabolic functions (*Meng et al., 2019*). On the other hand, some microbes can cause disease and further damage hosts (*Pirofski & Casadevall, 2020*). For example, *Pneumococcus* can cause pneumonia, sepsis and meningitis by colonizing the human upper respiratory tract (*Morimura et al., 2021*). Moreover, tetanus neurotoxin produced by *Clostridium tetani* inhibits neurotransmission, causing spastic paralysis in tetanus disease (*Masuyer, Conrad & Stenmark, 2017*). However, humans are fortunate to have discovered antibiotic drugs, which can inhibit or kill harmful microorganisms, becoming an effective method for combating microbe infections and diseases (*Pauter, Szultka-Młyńska & Buszewski, 2020*). Especially, antibiotic drugs can act on microbe through different mechanisms, thereby blocking their life cycle and controlling infection (*Baquero & Levin, 2021*). For example, antibiotics can intervene in key processes of cell wall synthesis (*Cho, Uehara & Bernhardt Thomas, 2014*), protein synthesis (*Hashemian, Farhadi & Ganjparvar, 2018*), and nucleic acid replication in bacteria, thereby inhibiting bacterial growth and reproduction (*Drlica & Zhao, 2021*). Unsurprisingly, the discovery and application of antibiotics is considered as a major breakthrough in the field of medicine (*Moser et al., 2019*). However, with the widespread use of antibiotics, microbe have also begun to develop resistance to antibiotics, resulting in a decrease in the ability of antibiotics to kill or inhibit microbe (*Bassegoda et al., 2018*). Therefore, it is urgent to develop new antibiotic drugs (*Bhardwaj et al., 2022*).

At present, the development of new drugs face two main challenges. The first challenge is the insufficient investment in drug research and drug development (*Fernandes & Martens, 2017*). In contrast, more research and investment are focused on drugs with long-term use, while the discovery and development of new drugs face relatively few resources and attention (*Martens & Demain, 2017*). Another challenge is the resistance of microbe. The growing challenge of antibiotic resistance in treating diseases caused by microbes susceptible to antibiotics diminishes the efficacy of current antibiotic drugs. This phenomenon makes it increasingly difficult to effectively address infections caused by susceptible microorganisms (*Eisenreich et al., 2022*). For example, patience infected with Human Immunodeficiency Virus (HIV) who have been undergoing long-term treatment with antiretroviral drugs (ARV) may generate drug-resistant virus strains, reducing sensitivity to conventional antiretroviral drugs (*Capetti & Rizzardini, 2019*). Therefore, there is an urgent need to develop new antiviral drugs to treat patients with HIV infection (*Lingappa, Lingappa & Reed, 2021*). Moreover, multidrug-resistant tuberculosis (MDR-TB) and extensively drug-resistant tuberculosis (XDR-TB) are two forms of resistance of bacteria to conventional TB drugs. These strains have developed resistance to many of the

drugs commonly used to treat TB, making treatment difficult (*Seung, Keshavjee & Rich, 2015*). Antimicrobial drug resistance poses a great threat to human beings in the word. It is estimated that antimicrobial resistance causes 700,000 deaths annually, and this number will rise to 10 million annually after 2050 (*Asare et al., 2022*). Currently, drug combination therapy and drug repurposing are effective approaches to combat antimicrobial resistance (*Foletto et al., 2021*; *Liu et al., 2021*). In particular, drug repurposing, which utilizes existing drugs to treat different diseases, is a safe and successful way to accelerate the discovery and development of new drug (*Konreddy et al., 2019*). It is worth mentioning that identification of microbes-associated drugs can facilitate the development of drug combination and drug repositioning. Therefore, developing effective methods to discovery potential microbe-drug associations is urgently needed.

Currently, with the continuous increase of available biological data, computational methods have emerged in predicting potential microbe-drug associations. Using computational models to predict drugs associated with microorganisms can significantly reduce the time and cost required for traditional wet experiments. The currently developed microbe-drug association prediction models can be broadly classified into two categories. The first type of prediction models is network-based models. For example, *Long & Luo (2021)* proposed a novel Heterogeneous Network Embedding Representation framework for Microbe-Drug Association prediction (HNERMDA). They constructed a heterogeneous network based on microbe-microbe associations, drug-drug associations, and known microbe-drug associations. To capture different semantic features hidden in diverse networks, the metapath2vec algorithm was used to learn the embedded representations of microbes and drugs based on heterogeneous networks. Furthermore, a biased network projection recommendation algorithm was implemented to identify new microbe-drug associations by assigning different bias weights to microbes and drugs. Moreover, *Zhu et al. (2019)* developed a new computational model of the KATZ measure for predicting novel human microbe-drug associations (HMDAKATZ). In the model, they first constructed a microbe-drug heterogeneous network by integrating multi-source data. Subsequently, based on the constructed heterogeneous network, they used the length-based algorithm of KATZ to predict potential microbe-drug associations by comprehensively considering paths of various lengths between microbe and drug.

The second type of prediction models is machine learning-based models. For example, *Long et al. (2020a)* proposed a graph convolutional network (GCN) based framework for predicting human Microbe-Drug Associations named GCNMDA to predict potential microbe-drug associations. First, they constructed a heterogeneous network by integrating drug similarity network, microbe similarity network and microbe-drug association network. Next, the restart random walk algorithm was applied to process both the microbe similarity network and the drug similarity network, resulting in the derivation of a microbe feature matrix and a drug feature matrix. Then, they used GCN to learn node embeddings based on the heterogeneous networks and the new feature matrices. Furthermore, within the hidden layer of GCN, a conditional random field (CRF) was introduced to augment the learning efficacy of node representation, ensuring that analogous nodes (*i.e.,* microbes or drugs) exhibit comparable representations. To accurately aggregate representations of

communities, they applied the attention mechanism in the CRF. Finally, they established a reconstructed bipartite graph for the prediction of microbe-drug associations. Besides, based on the research foundation of GCNMDA, *Long et al. (2020b)* further proposed a novel Ensemble framework of Graph Attention networks with a hierarchical attention mechanism for Microbe-Drug Association prediction (EGATMDA). Firstly, three distinct networks (graphs) were established based on microbe, drug, and disease factors, resulting in a microbe-drug bipartite network, a microbe-drug heterogeneous network, and a microbe-disease-drug heterogeneous network. Secondly, microbe feature matrix and drug integrated matrix are constructed based on drug structure similarity, drug Gaussian kernel similarity and microbe gene sequence similarity. Thirdly, they employed graph convolutional networks (GCN) and graph attention networks (GAT) to learn node embedding representations for each microbe and drug node from the feature matrix and each input microbe-drug network. In their methodology, each node, representing both microbe and drug, holds diverse semantic information within different graphs. To effectively integrate this information and alleviate noise originating from diverse graphs, they propose the utilization of a graph-level attention mechanism. Additionally, *Deng et al. (2022)* proposed a computational model called Graph2MDA that utilizes variational graph autoencoders (VGAE) and deep neural networks (DNN) to predict potential microbe-drug associations. First, they constructed a multimodal property graph that included drug structural similarity, drug Gaussian kernel similarity, microbial functional similarity, and microbial sequence similarity. Then, the latent representation of the nodes (*i.e.*, microbes or drugs) is learned using VGAE. Finally, they used a deep neural network classifier to predict potential microbe-drug associations based on the node embeddings learned by VGAE. To fully harness the informative potential embedded in similar (neighbor) characteristics of drugs or microbes and unveil the latent probability distribution within established microbe-drug associations, *Cheng et al. (2022)* implemented a two-step methodology. Initially, they utilized neighborhood integration (NI) to generate a score matrix for potential microbe-drug associations. This involved employing diverse thresholds to identify comparable neighbors for drugs or microbes. Subsequently, the authors applied the Restricted Boltzmann Machine (RBM) to derive another score matrix for potential microbe-drug associations, leveraging the contrastive divergence algorithm and sigmoid function. To enhance predictive accuracy, an ensemble learning approach was employed to integrate these two score matrices.

In this article, based on the known microbe-drug associations, drug similarity and microbe similarity, we proposed a new computational model of convolutional neural network with Bernoulli Random Forest for Microbe-Drug Association prediction (CNNBRFMDA) based on CNN and BRF (*Deepthi, Jereesh & Liu, 2021*; *Wang et al., 2018*). Firstly, drug chemical structure similarity and drug side effect similarity were combined into integrated drug similarity. Similarly, Integrated microbe similarity was constructed through microbe sequence similarity. Subsequently, feature vector for each microbe-drug pair can be constructed by concatenating integrated drug similarity and integrated microbe similarity. Next, CNN is utilized to extract latent feature representations from the feature vectors of all microbe-drug pairs and low-dimensional feature vectors for all microbe-drug
**Table 1   The statistics for each microbe-drug association dataset.**

| Datasets | Drugs | Microbes | Associations |
|----------|-------|----------|--------------|
| abiofilm | 1,720 | 140 | 2,884 |
| MDAD | 1,373 | 173 | 2,470 |

pairs can be generated. Finally, we employed BRF to infer potential associations between microbes and drugs. In addition, we implemented five-fold cross-validation to evaluate the capabilities of CNNBRFMDA based on the MDAD and abiofilm datasets, respectively. The results showed that the mean AUCs and standard deviations of the 5-fold cross-validation were 0.9017 +/- 0.0032 for the MDAD dataset and 0.9146 +/- 0.0041 for the abiofilm dataset. Two different case studies were conducted to evaluate the performance of CNNBRFMDA. The results showed that 41 out of the top 50 microbes predicted for ciprofloxacin were confirmed by literatures. Another case studies showed that 38 out of the top 50 predicted microbes for moxifloxacin were confirmed by searching the published literatures.

## MATERIALS & METHODS

### Microbe-drug association

In this article, we introduced two datasets of known microbe-drug associations, separately for abiofilm (*Rajput et al., 2018*), Microbe-Drug Association Database (MDAD) (*Sun et al., 2018*). The abiofilm dataset records 2,884 known microbe-drug associations between 1,720 drugs and 140 microbes. Moreover, MDAD dataset collected 2,470 known microbe-drug associations between 1,373 drugs and 173 microbes. The statistics of the two datasets mentioned above are shown in Table 1. Furthermore, the adjacency matrix $A \in R^{nd \times nm}$ was used to represent the microbe-drug association information, where $nd$ means the number of drugs and $nm$ means the number of microbes. The value of entity $A_{ij}$ is set to 1 when the drug $d_i$ was associated with the microbe $m_j$. Otherwise, the value is set to 0.

$$A_{ij} = \begin{cases} 1, \text{if durg } d_i \text{ is associated with microbe } m_j \\ 0, \text{otherwise} \end{cases} \tag{1}$$

### Drug side effect similarity

The SIDER (*Kuhn et al., 2010*) database provides extensive information about drug side effects, and we downloaded the side effects of the drugs used in the this study from SIDER. Based on the view that the more side effects shared by two drugs, the more similar the two drugs. The set of side effects associated with drug $d_i$ is defined as $N(i)$. We defined the similarity between two drugs as 0 when there were no common side effects between the two drugs. The entity $SS1(i,j)$ denoted the similarity of side effects between drug $d_i$ and drug $d_j$. Jaccard (*Gottlieb et al., 2011*) was utilized to calculate drug side effect similarity Eq. (2) showed as following.

$$SS1 = \text{Jaccard score} = \frac{|N(i) \cap N(j)|}{|N(i) \cup N(j)|}. \tag{2}$$

## Drug chemical structure similarity

SIMCOMP (http://www.genome.jp/tools/simcomp/) is a graph-based chemical structure comparison method that can be used to obtain the chemical structure similarity of different drugs. SIMCOMP is used to find maximal cliques in association graphs and furthermore to discover isomorphic maximal common subgraphs (*Hattori et al., 2003*). In the data pre-processing of this article, SIMCOMP2 quantifies the similarity between drugs by comparing their chemical structures. Specifically, we set the cutoff score for drug structure similarity at 0.5. The drug chemical structural similarity between drug $d_i$ and drug $d_j$ was represented by $SS2$.

## Microbe sequence similarity

In this paper, we obtained two microbe sequence similarity based on two different known microbe-drug association datasets, respectively. For microbes in the MDAD dataset or the abiofilm dataset, microbe sequence similarity obtained by MAFFT and BioEdit calculations. MAFFT introduces the approximate distance calculation algorithm and the fast Fourier alignment algorithm to achieve multiple sequence matching. The sequence similarity matrix can be calculated by using the sequence identification matrix function in BioEdit (*Tippmann, 2004*). Based on the idea that the more sequences two microbes have in common, the more similar they are to each other. Hence, when two microbes have no sequence in common, the value of the microbe sequence similarity score between the two microbes is equal to 0. In this article, the microbe sequence similarity matrix was represented by the $MV$ matrix, as in which $MV(m_i, m_j)$ denotes the sequence similarity value between microbe $m_i$ and microbe $m_j$.

## Integrated similarity for microbe and drug

To fully leverage the raw information of drugs and microbes, we construct the integrated similarity for microbes using unprocessed microbe sequence similarity. For drugs, we aggregate the drug chemical structure similarity and drug side effect similarity and compute their average to obtain the integrated similarity for drugs. CNNBRF demonstrates outstanding performance by thoroughly exploiting the original data.

Moreover, to ensure consistency in data processing, we exclude the parts of models like NIRBMMDA, LAGCNMDA, and RFMDA that utilize Gaussian kernel similarity. We observe a significant decrease in performance when these parts are removed, while our model maintains its robustness. This resilience is one of the strengths of our model.

## CNNBRFMDA

In this article, we propose a new computational model named CNNBRFMDA to predict potential microbe-drug associations. The flowchart of CNNBRFMDA is shown in Fig. 1. The proposed method consists of three main parts: Data preparation, CNN for latent feature extraction and BRF for predicting potential microbe-drug association.

## Data preparation

Data preparation is the first step of the CNNBRFMDA algorithm, aimed at representing microbe-drug pairs as feature vectors. As previously mentioned, in our study, we
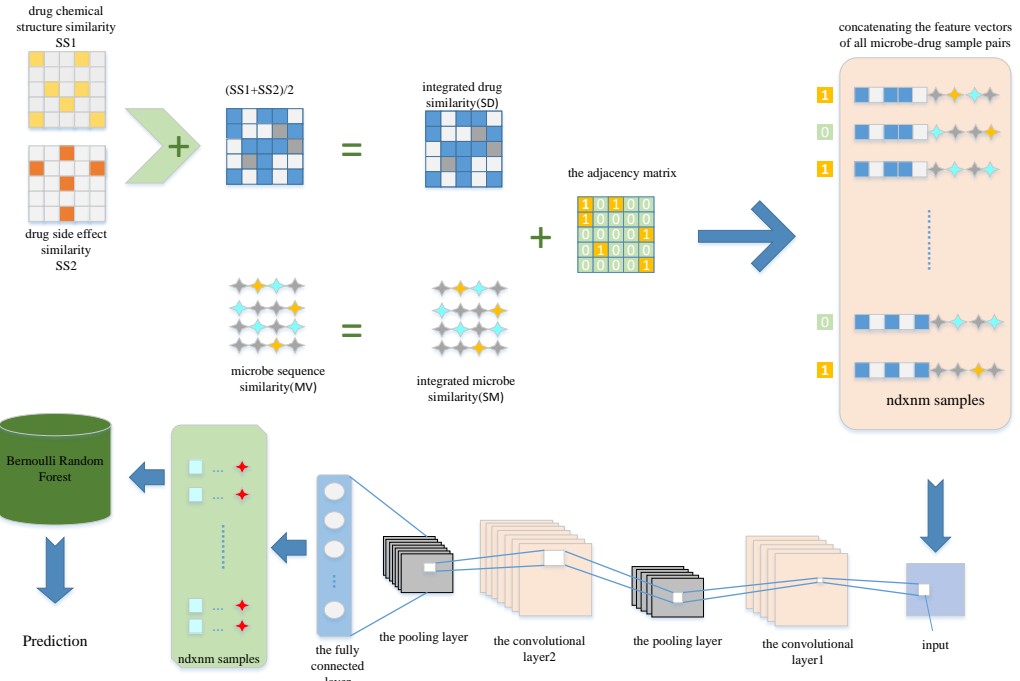

**Figure 1** **The flowchart of CNNBRFMDA.** The proposed method consists of three main parts: Data preparation, Convolutional Neural Network (CNN) for latent feature extraction and Bernoulli Random Forest (BRF) for predicting potential microbe-drug association.

constructed the adjacency matrix A of known microbe-drug association, the integrated drug similarity matrix $SD(nd \times nd)$ and the integrated microbe similarity matrix $SM(nm \times nm)$. We extracted $nm$ and $nd$ features separately for microbe and drugs from these matrices, respectively. By concatenating the feature vectors of the investigated microbes and drugs, we obtained $nm + nd$ features for each microbe-drug pair. For any one of the two datasets, each drug is comprised of $nd$ features $(f_1, f_2, \ldots, f_x \ldots, f_{nd})$, where the $x - th$ feature $f_x$, represents the integrated similarity between the investigated drug and the $x - th$ drug. Therefore, drug $d_u$ can be represented by $nd$-dimensional vector $FeV(d_u)$, which can be described as Eq. (3):

$$FeV(d_u) = (f(u)_1, f(u)_2, \ldots, f(u)_x, \ldots, f(u)_{nd}) \tag{3}$$

where the $x - th$ element $f(u)_x$ of $FeV(d_u)$ is the integrated similarity between drug $d_x$ and drug $d_u$. Similarity, we denoted the $nm$-dimensional vector $FeV(m_v)$ for microbe $m_v$ as Eq. (4):

$$FeV(m_v) = (f(v)_1, f(v)_2, \ldots, f(v)_y, \ldots, f(v)_{nm}) \tag{4}$$

where the $y - th$ element $f(v)_y$ of $FeV(m_v)$ is the integrated similarity between microbe $m_y$ and $m_v$.

Subsequently, by concatenating the feature vectors of $FeV(m_v)$ and $FeV(d_u)$, we obtained $nm + nd$ features for each microbe-drug pair. As a result, there were $nm \times nd$ samples and

the length of each sample was $nm + nd$, each sample corresponds to a microbe-drug pair. Next, we set the labels of the known microbe-drug association to 1, forming the positive sample set $P$. Simultaneously, negative samples were selected from the unlabeled microbe-drug pairs, and their labels were set as 0 to create a negative sample set $N_{CNN}$, ensuring a balanced 1:1 ratio of positive sample set and negative sample set. This method is feasible due to the low ratio of $\frac{\text{(the number of P)}}{nm \times nd - \text{(the number of P)}}$. After compiling $P$ and $N$, the next step was combined them into a training set to construct low- dimension feature vectors.

## Convolutional neural network for latent feature extraction

CNN is a deep learning algorithm (*LeCun et al., 1989*), the three basic layers of CNN are the convolutional layer, the pooling layer (the subsampling layer), and the fully connected layer. The convolutional layer obtains different features of the input data by learning filters, and then the pooling layer is used to reduce the feature dimension, enhance the translation invariance and robustness of the model, and finally classify in the fully connected layer. In the CNNBRFMDA model, we utilized CNN to extract complex features from microbe-drug feature vectors in a deep-learning framework. Specifically, we perform multiple convolution operations on input samples using different kernels to generate activation maps. The activation map $Q_l$ at layer $l$ can be described as:

$$Q_l = \sigma(Q_{l-1} \odot W_l + b_l) \tag{5}$$

where $\sigma(k)$ denotes the activation function, $W_l$ the convolution kernel at layer $l$, $b_l$ the offset vector, and $\odot$ the convolution operation. Subsampling layers are used to compress the feature vectors of microbe-drug pairs and minimize overfitting. The formula for the subsampling layer $Q_l$ can be expressed as:

$$Q_l = Subsampling(Q_{l-1}). \tag{6}$$

In order to preserve the most salient features of each filtered region, we employ max-pooling in the subsampling layer. During the training process, CNN was optimized to minimize the network's loss function. The training data was fed into CNN, enabling it to capture salient features from the input feature vectors of microbe-drug pairs. We conducted several experiments to identify the optimal hyperparameters of the CNN and obtain the best predictive model. For the convolutional operation, we incorporated 16 filters with a size of $1 \times 16$ in the convolutional layer, based on the configuration used in *Deepthi, Jereesh & Liu (2021)*, which has been shown to perform effectively for similar tasks. In the subsampling layer, we established a filter size of $1 \times 2$. In our model, At the convolution and fully connected layers, we employed the rectified linear unit (ReLU (*Nair & Hinton, 2010*)) as the activation function. Additionally, the sigmoid function was used at the output layer. The model was implemented with the binary cross-entropy error function and optimized using Adam. Furthermore, to avoid overfitting, two dropout layer (*Srivastava et al., 2014*) are added between the convolutional and fully connected layers. Finally, we retrieved the weight data learned after multiple convolution and pooling operations. Ultimately, we updated the feature vectors of all microbe-drug pairs using the weight data learned from CNN. Specifically, we used the output data of the second-to-last layer (fully connected

layer) as the low-dimension feature vectors for all microbe-drug pairs. Here, we reduced the dimensionality to $\frac{nd+nm}{2}$.

The model prevents overfitting through several strategies, including the use of dropout layers that randomly deactivate neurons to reduce model dependency on any specific set of features, as well as employing five-fold cross-validation. This cross-validation technique helps ensure that the model's performance is robust across different subsets of the data, further mitigating the risk of overfitting.

It is worth mentioning that the model consists of only three layers with trainable parameters: one convolutional layer and two fully connected layers. This streamlined architecture significantly reduces computational complexity, decreases memory usage, and accelerates the training process, making it exceptionally efficient for deployment in environments with limited computational resources. Furthermore, the simplicity of having fewer trainable layers reduces the risk of overfitting and streamlines the model tuning and optimization process, ensuring robust performance across diverse datasets. Consequently, the model remains both practical and effective for applications that demand rapid processing with reliable accuracy.

## Bernoulli random forest based prediction

As a learning algorithm, random forest (RF) is widely used in various prediction tasks and has achieved notable performance (*Chen et al., 2018*; *Wang et al., 2019*). This technique involves three key processes: random bootstrap sampling, random attribute bagging, and tree node splitting.

First, random bootstrap sampling is a method of sampling with replacement. For each tree, the random forest algorithm selects a subset of the training data, known as a "bootstrap sample", which may contain duplicate data points. Second, random attribute bagging involves selecting a random subset of features for each node split rather than using all available features. Finally, tree node splitting is the core step in constructing decision trees. At each node, the algorithm selects the optimal feature and split point based on criteria such as Gini impurity or information gain (see Fig. S1).

However, RF cannot simultaneously account for both theoretical consistency and empirical reliability. The BRF, a variant of RF introduced in this article, demonstrates good experimental performance while maintaining theoretical consistency. Compared to RF, BRF achieves theoretical consistency while retaining strong performance. The key differences lie in their approaches to random bootstrap sampling, random attribute bagging, and tree node splitting

BRF divides the training data into a structural part and an estimation part. The structural part is used to construct the tree's shape, while the estimation part is reserved exclusively for sample prediction and not used in tree construction. This contrasts with RF, where the entire bootstrap sample is used for both tree construction and prediction.

In the BRF, the feature selection for node splitting is not conducted by merely randomly selecting a subset of features. Instead, Bernoulli trials are employed to decide whether to select a single feature or a subset of features. This method introduces controlled randomness into the feature selection process, thereby enhancing model robustness. As illustrated in

Fig. S2, event B1 chooses a value from 0 or 1 with probability $p1$, where B1 follows a Bernoulli distribution. Here, $p1$ corresponds to choosing one candidate feature, while $1-p1$ corresponds to selecting $\sqrt{D}$ candidate features from the $D$ features in training data. In contrast, traditional RF selects features purely at random without such control. After selecting features and samples, RF deterministically splits tree nodes using conventional decision tree algorithms. Conversely, BRF utilizes Bernoulli trials to determine whether to select the splitting point based on the impurity criterion or random sampling. This decision process is controlled by B2: Bernoulli trial controlled tree node splitting, where $p2$ represents random sampling, and $1-p2$ represents splitting based on the impurity criterion. This additional layer of randomness adds complexity to the tree structure. These three differences collectively ensure that as data size grows indefinitely large, BRF converges to the optimal solution, thereby achieving consistency.

# RESULTS

## Performance evaluation

We conducted crossover experiments to assess the performance of CNNBRFMDA based on two representative datasets: MDAD and abiofilm. Each dataset was trained separately, with the abiofilm dataset containing 2,884 known associations between 1,720 drugs and 140 microbes, the MDAD dataset containing 2,470 known associations between 1,373 drugs and 173 microbes. To validate the predictive performance of CNNBRFMDA, 5-fold cross-validation was performed. All known microbe-drug associations were randomly divided into five equally sized subsets, with each subset being used as the test set in rotation. In comparison, the remaining four subsets were used as the training set. CNNBRFMDA was used to score all unlabeled microbe-drug pairs and test samples, and the scores of each test sample were compared with the scores of all unlabeled pairs to determine their ranking. Additionally, by setting the association values for the microbe-drug pairs to be predicted as 0, we simulated a scenario where the model predicts new, unseen microbes or drugs that are not present in the current datasets. This approach mirrors real-world conditions, where the model would encounter novel entities, providing a realistic estimate of its predictive capabilities in such situations.

Based on the CNN parameters derived from *Deepthi, Jereesh & Liu (2021)*, we conducted a comparative analysis of CNNBRFMDA against five models: CNNRFMDA, BRFMDA (*Wang et al., 2018*), NIRBMMDA (*Cheng et al., 2022*), LAGCNMDA (*Qu et al., 2024*), RFMDA (*Chen et al., 2018*), employing a five-fold cross-validation.

For MDAD dataset, as shown in Fig. S3, CNNBRFMDA obtained an AUC of 0.9074, which was superior to CNNRFMDA (0.9043), BRFMDA (0.8873), NIRBMMDA (0.8589), LAGCNMDA (0.8241), RFMDA (0.7834).

For abiofilm dataset, as shown in Fig. S4, CNNBRFMDA obtained an AUC of 0.9221, which was superior to CNNRFMDA (0.9079), BRFMDA (0.8816), NIRBMMDA (0.8660), LAGCNMDA (0.8425), RFMDA (0.8313).

The corresponding Precision-Recall (PR) curves for the CNNBRF model are presented in Figs. S5 and S6, based on the MDAD and abiofilm datasets, respectively. We observed

that CNNBRF significantly outperforms both the random forest model and the baseline random model in terms of precision and recall. This indicates that the CNNBRF model is more effective at identifying true microbe-drug associations when processing the MDAD and abiofilm datasets. From the graphs, it is evident that the CNNBRF model maintains a high level of precision across most levels of recall, demonstrating that the model can retain high result reliability without sacrificing significant recall. In contrast, the random forest model shows a more pronounced decline in precision at higher recall levels. Moreover, the overall precision performance of CNNBRF is stable and does not rely on optimization for any specific threshold, indicating that the model performs consistently under varying conditions.

Taking the MDAD dataset as an example, the CNN model initially reduces the dimensionality of the input data, resulting in 773-dimensional feature vectors. These vectors are then input into the BRF, where they are used to construct decision trees. To improve interpretability, we have focused on the features utilized for splitting nodes within these trees. We have generated correlation coefficient plots of these features, as shown in Fig. S7. This visualization demonstrates low inter-feature correlation, suggesting that the features used for splitting in the BRF possess good independence. This independence is key to understanding the model's robust predictive performance.

To mitigate bias resulting from the random partitioning of known microbe-drug associations, the 5-fold cross-validation process was repeated 10 times. As a result, for the two datasets of MDAD, and abiofilm, CNNBRFMDA acquired the average AUC and standard deviation of $0.9017+/-0.0032$ and $0.9146+/-0.0041$, respectively, which were higher than those of five previous models. Additionally, we integrated logistic regression and k-nearest neighbors into our comparative analysis to further validate the performance of our model against these well-established techniques (see Tables S1, S2). It is worth mentioning that all prediction models underwent evaluation against CNNBRFMDA, utilizing the same dataset in 5-fold cross-validation.

To validate the efficacy of CNNBRF across different biological contexts, we employed the DrugVirus dataset (Andersen et al., 2020), which contains 933 associations between 175 drugs and 95 viruses. After conducting ten times five-fold cross-validation, we observed a performance metric of $0.7963 \pm 0.0017$. This result is lower compared to the performances achieved with the MDAD and abiofilm datasets. We believe this underperformance can largely be attributed to the smaller size of the DrugVirus dataset, which limits the model's ability to generalize as effectively as it does with larger datasets. The results highlight the challenges associated with smaller datasets and emphasize the need for further refinement of CNNBRFMDA to enhance its robustness and accuracy. Improving model performance across various data scales and biological contexts will be a key focus of our future work.

To investigate the impact of different epochs on model performance, we conducted performance tests at epochs 10, 20, 50, and 100 on the MDAD and abiofilm datasets, as depicted in (see Tables S3, S4). We found that at epoch 20, the performance achieved by the training was optimally balanced with computational resource consumption.

## Case studies

In our study, we conducted two case studies using the CNNBRFMDA model, building upon previous research to further validate its predictive capabilities. These case studies were informed by the methodologies and findings of earlier studies, as referenced in *Cheng et al. (2022)*, *Fan, Wang & Zhu (2023)*, *Li et al. (2023a)*, *Long et al. (2020a)*, *Tian et al. (2023)*. The first case study focused on predicting potential microbe associations of the drug ciprofloxacin based on the abiofilm dataset using CNNBRFMDA. Ciprofloxacin is one of a new generation of fluorinated quinolones. The results of clinical trials with orally and intravenously administered ciprofloxacin have confirmed its potential for use in a wide range of infections (*Campoli-Richards et al., 1988*). We employed CNNBRFMDA to predict ciprofloxacin-related microbes, subsequently ranking the top 50 potential ciprofloxacin-related microbes. These predictions were then validated by reviewing the relevant literature on PubMed. The results showed that 41 out of the top 50 predicted ciprofloxacin-related microbes were confirmed through literature review (see Table S5). For instance, the top-ranked ciprofloxacin-related microbe is *Escherichia coli*, a Gram-negative bacterium in the family Enterobacteriaceae. It can cause various disease syndromes such as different diarrheal diseases, wound infections, meningitis, septicemia, and atherosclerosis (*Olsvik et al., 1991*). *Jakobsen, Lundberg & Frimodt-Møller (2020)* found that ciprofloxacin was highly effective in eliminating susceptible *Escherichia coli* in urine and kidney tissues. This specific effectiveness is crucial, given the increasing antibiotic resistance among *E. coli* strains, which complicates the management of such infections. The activity of ciprofloxacin against *E. coli* not only highlights the importance of this antibiotic in the arsenal against Gram-negative bacterial infections but also emphasizes the need for targeted antibiotic therapy. This approach minimizes the misuse of broad-spectrum antibiotics and helps preserve the efficacy of these drugs, which is essential for slowing the development of resistance.

Shigellosis, predicted to be related to ciprofloxacin and ranked 13th, is a major global cause of dysentery (*Thompson, Duy & Baker, 2015*). It often manifests with symptoms such as diarrhea, abdominal pain, fever, and vomiting, and in severe cases, it can lead to dehydration and other complications. *Gharpure et al. (2022)* investigated an outbreak of multidrug-resistant *Shigella sonnei* infections in a retirement community and found that the isolated strains were susceptible to ciprofloxacin. The effectiveness of ciprofloxacin against drug-resistant *Shigella sonnei* provides a reliable treatment pathway, particularly in outbreak scenarios where a quick and effective response is crucial. This underscores the importance of maintaining ciprofloxacin as a viable treatment option in the face of rising antibiotic resistance, ensuring that healthcare providers can effectively manage outbreaks and prevent widespread transmission and morbidity.

Additionally, *Salmonella enterica* was predicted to be related to ciprofloxacin and ranked 14th. *Salmonella enterica*, a Gram-negative rod-shaped bacterium, is commonly transmitted through contaminated water and food and can cause several different disease syndromes, with the most common being gastroenteritis, followed by bacteremia and typhoid fever (*Han et al., 2024*). *Li et al. (2023b)* discovered that ciprofloxacin can be used in the treatment of *Salmonella enterica* serovar Typhimurium infection. The model's

identification of ciprofloxacin as effective against *Salmonella enterica* serovar Typhimurium is particularly important given the bacterium's role in significant public health issues like gastroenteritis and typhoid fever. Ciprofloxacin's efficacy highlights its importance not only in treating common infections but also severe systemic infections caused by Salmonella, providing a reliable treatment option in the face of increasing antibiotic resistance. This underscores the value of targeted antibiotic therapy, allowing for more precise treatment strategies and helping to preserve the effectiveness of essential drugs against resistant strains.

In the second case study, we aimed to verify the predictive ability of CNNBRFMDA on microbe potentially associated with new drugs. To achieve this object, we removed all known associations of the drug moxifloxacin in the MDAD dataset and utilized the processed MDAD dataset to predict potential associated microbe of the new drug moxifloxacin using CNNBRFMDA. The results showed that 38 out of the top 50 predicted moxifloxacin-related microbes were confirmed through literature review (see Table S6). For example, with the highest correlation score to moxifloxacin, *Haemophilus influenzae*, a pathogenic bacterium causing respiratory and otolaryngological infections (*Honda et al., 2020*), has been experimentally confirmed by Honda et al. to be susceptible to moxifloxacin treatment. This finding is particularly important for guiding antibiotic stewardship and treatment strategies. By identifying *Haemophilus influenzae* as highly responsive to moxifloxacin, healthcare providers can more accurately prescribe treatments that are likely to be effective, reducing the need for broader spectrum antibiotics that may contribute to the development of further resistance. Additionally, this outcome highlights the potential of moxifloxacin as a valuable antibiotic in managing infections that are difficult to treat due to either the location or the nature of the bacterial resistance. Additionally, *Candida albicans*, which ranks second in correlation to moxifloxacin, is a commonly encountered fungal pathogen associated with diseases ranging from superficial mucosal complaints to life-threatening systemic disorders (*Wang, 2015*). *Jadhav et al. (2017)* have demonstrated the potential impact of moxifloxacin on the growth, morphogenesis, and abiofilm formation of *Candida albicans*, with minimal likelihood of *Candida albicans* developing resistance to moxifloxacin. This efficacy against *Candida albicans* is particularly significant given the pathogen's association with a range of diseases from mild mucosal ailments to severe systemic infections that can be life-threatening. The ability of moxifloxacin to inhibit biofilm formation—a key factor in the virulence and drug resistance of Candida infections—further enhances its potential utility in clinical settings where fungal infections are prevalent or complicated by biofilms. The implications of these findings are broad, not only for the treatment of fungal infections but also for the potential development of moxifloxacin as a multi-use antimicrobial agent. This could lead to its strategic use in cases where fungal and bacterial co-infections are suspected, thus simplifying treatment regimes and potentially reducing the need for combination therapies that may have higher toxicities or drug interaction risks. *Staphylococcus aureus*, which ranks sixth in correlation to moxifloxacin, is a clinically important pathogen causing a wide range of human infections, from minor skin infections to severe tissue infection (*Ahmad-Mansour et al., 2021*). *Ince, Zhang & Hooper (2003)* have confirmed that moxifloxacin exhibits enhanced potency against *Staphylococcus aureus*,

lower propensity to select for resistant mutants, and higher bactericidal activity against highly resistant strains. Moxifloxacin's ability to effectively target *Staphylococcus aureus* with a lower propensity for developing resistance is particularly significant in the context of increasing antibiotic resistance worldwide. This characteristic makes moxifloxacin an important option in the antibiotic arsenal, especially in settings where drug resistance is prevalent, and traditional treatments fail. Its higher bactericidal activity is also critical for ensuring effective eradication of infections, minimizing the risk of recurrence and the spread of infection within healthcare settings. Moreover, the broad efficacy of moxifloxacin against *Staphylococcus aureus* enhances its utility not just for typical bacterial infections but also for more complicated clinical scenarios where the bacterial load and resistance factors complicate treatment strategies. This versatility can lead to better clinical outcomes and may reduce the duration and complexity of antibiotic therapy needed to manage infections effectively.

## DISCUSSION

Because the problem of microbe resistance to drugs is intensifying and the development cycle of new drugs is prolonged, more and more researchers have begun to pay attention to computational models for the prediction of microbial-drug associations (*Liang et al., 2024*; *Xuan et al., 2024*; *Zhao et al., 2024*; *Zhu et al., 2024*). In this study, we developed a computational model CNNBRF for microbe-drug association prediction. First, we leverage known microbe-drug associations to construct a positive sample set. At the same time, in order to maintain a 1:1 ratio between the positive sample set and the negative sample set, we randomly choose an equal number of samples from unknown microbe-drug pairs to form the negative sample set. Second, feature vectors are constructed for each microbe-drug pair by utilizing the integrated microbe similarity and integrated drug similarity. Then, a convolutional neural network is employed to extract latent features and construct low-dimensional feature vectors for each microbe-drug pair. Finally, we used the BRF model to predict potential microbe-drug associations. We performed 5-fold cross-validation to verify the accuracy of CNNBRFMDA. In addition, we conducted two types of case studies. The results of the extended analysis demonstrate the well performance of CNNBRFMDA in predicting potential microbe-drug associations.

To transition the findings from the CNNBRFMDA model into clinical or pharmaceutical trials, the following steps could be implemented: The first step involves rigorous validation of the model's predictions using independent laboratory tests to verify the predicted microbe-drug interactions. This could include *in vitro* testing to assess the efficacy and safety of the drug interactions identified by the model. Specifically, researchers can obtain microbe-drug association data of interest, similar to what we described in the Materials & Methods section of our study. They can then input this association data into the CNNBRFMDA model and use the scores generated for these microbe-drug pairs to determine whether to proceed with actual clinical or pharmaceutical trials. The model will provide valuable insights and serve as a reference for designing these trials, helping to prioritize interventions based on the predicted effectiveness and safety of the interactions.

Following the initial validation, the next steps would involve detailed planning and implementation of pilot studies. These preliminary studies are crucial as they allow for the assessment of the model's predictions in a controlled, smaller-scale clinical environment before proceeding to larger trials. During these pilot studies, the practical application of the CNNBRFMDA model's predictions can be observed firsthand, providing an opportunity to fine-tune the trial design based on real-world data and feedback.

Several key factors contribute to the reliable empirical performance and theoretical consistency of CNNBRFMDA. CNN serves as a powerful feature extractor, enabling automatic learning of high-quality features from raw input data. However, in regression tasks, CNN models tend to be complex and lack interpretability. Hence, we employ BRF to fulfill the regression task. BRF introduces Bernoulli distribution during the feature selection and split node selection process, allowing for the generation of both deterministic and uncertain tree structures. Through these operations, we gain insights into how the model makes regression predictions for microbe-drug association based on features. Compared to traditional RF, BRF exhibits less dependency on data while offering well performance while maintaining theoretical consistency. In conclusion, CNNBRFMDA demonstrates excellent performance in predicting potential microbe-drug associations, along with strong interpretability. CNNBRFMDA is applicable to drugs (microbes) that have not been proven to be related to microbes (drugs). Therefore, we can use CNNBRFMDA to predict potential microbe-drug association. Most importantly, predicting potential microbe-drug associations using the computational model is cost-effective, fast, and efficient, which can minimize clinical trial failures.

The method of randomly selecting samples from unlabeled microbe-drug pairs to construct the negative sample set, as employed in our study, can introduce biases that might affect the predictive accuracy of our model. This approach assumes that all unlabeled pairs are negative, which may not always be the case. Some of these unlabeled pairs could be true but undiscovered associations, leading to false negatives. This mislabeling can potentially skew the model's learning, making it biased towards false assumptions about non-associations. Such biases can reduce the generalizability of the model to new, unseen data and might limit its effectiveness in accurately predicting true microbe-drug interactions. In our study, to address data imbalance and improve the accuracy of model predictions, we adopted the Density-aware Stratified Sampling (DAS) method for reselecting negative samples. The core of this method is to consider the density information of samples for more targeted sampling, ensuring that the negative samples selected are representative in the feature space. The specific process is as follows: We first calculate the feature vectors for each unlabeled microbe-drug pair. Utilizing these feature vectors, we calculate the density for each sample, defined as the number of neighboring samples within a predefined radius. This step helps to identify samples across varying density regions of the feature space. After calculating densities, we sort all samples based on their density values. Instead of selecting a fixed number of high-density samples, we employ systematic sampling to ensure a diversity of densities is included. Systematic sampling involves selecting samples from the sorted list at fixed intervals, which helps capture a representative sample across the entire spectrum of density distributions, from low to high. This approach ensures that

the negative samples selected are not only numerically balanced with positive samples but also represent a variety of density distributions, closely mirroring real-world data distributions. By employing systematic sampling, we minimize bias in sample selection, thus enhancing the model's generalization ability to new, unseen data. The final results of the CNNBRF-DAS model, after conducting five-fold cross-validation ten times on the MDAD dataset, showed the mean AUC of 0.9062 with a standard deviation of $\pm$0.0034. Similarly, on the abiofilm dataset, the model achieved the mean AUC of 0.9177 with a standard deviation of $\pm$0.0023 through the same validation process.

In addition, we need to expand known microbe-drug associations. Considering the algorithmic characteristics of CNNBRF, we are optimistic about achieving significant breakthroughs in predicting potential associations with continued data expansion and further enhancement of our tools. Looking ahead, an exciting avenue for our research lies in extending our model to predict resistance trends. This will necessitate integrating longitudinal data to capture the dynamics of microbial resistance over time, including microbial genomic and phenotypic data alongside information on drug exposure. Although comprehensive data sets required for such predictions are currently scarce, advances in bioinformatics bolster our confidence that developing models capable of predicting resistance trends is within reach.

## CONCLUSIONS

We proposed a novel model named CNNBRFMDA to predict potential microbe-drug associations. In this model, we employed a convolutional neural network to extract features from microbe-drug pairs. Considering that the high dimensionality of the feature vectors for microbe-drug pairs might contain a lot of unnecessary information, we applied dimensionality reduction to the feature vectors. Subsequently, we used an innovative method, BRF, to predict potential microbe-drug associations. Additionally, we evaluated the performance of CNNBRFMDA on the MDAD and abiofilm datasets using five-fold cross-validation, and plotted the Precision-Recall curves. The results showed that CNNBRFMDA outperformed other models such as CNNNRFMDA, BRFMDA, NIRBMMDA, LAGCNMDA, and RFMDA. Furthermore, two case studies involving Ciprofloxacin and moxifloxacin demonstrated that CNNBRFMDA is an effective predictive model. Despite its promising predictive performance in case studies, CNNBRFMDA still has room for improvement. Firstly, the predictive accuracy of the model could be enhanced by collecting more features of microbes and drugs to enrich the similarity features between them. Secondly, using deeper convolutional neural networks could further extract the deep features of microbe-drug pairs.

## ACKNOWLEDGEMENTS

The authors acknowledge the use of ChatGPT (OpenAI) for assistance in language editing and improving the readability of the manuscript. We appreciate their valuable contribution to enhancing the quality of our writing. We also express our sincere gratitude to the editor

and reviewers for their valuable comments and suggestions, which significantly contributed to the improvement of this manuscript.

### Funding

Jia Qu was supported by "Natural Science Foundation of Jiangsu Province under Grant BK20220621". Qinggang Bu was supported by "Natural Science Foundation of Jiangsu Province under Grant BK20230621". Zekang Bian was supported by "National Natural Science Foundation of China under Grant 62306126, Wuxi Science and Technology Development Fund Project under Grant K20231006." The funders had no role in study design, data collection and analysis, decision to publish, or preparation of the manuscript.

### Grant Disclosures

The following grant information was disclosed by the authors:
Natural Science Foundation of Jiangsu Province: BK20220621, BK20230621.
National Natural Science Foundation of China: 62306126.
Wuxi Science and Technology Development Fund Project: K20231006.

### Competing Interests

The authors declare there are no competing interests.

### Author Contributions

- Zihao Song conceived and designed the experiments, performed the experiments, analyzed the data, prepared figures and/or tables, authored or reviewed drafts of the article, and approved the final draft.
- Qingnuo Li conceived and designed the experiments, performed the experiments, analyzed the data, authored or reviewed drafts of the article, and approved the final draft.
- Jincheng Zhao performed the experiments, prepared figures and/or tables, and approved the final draft.
- Qinggang Bu performed the experiments, authored or reviewed drafts of the article, and approved the final draft.
- Zekang Bian performed the experiments, analyzed the data, prepared figures and/or tables, authored or reviewed drafts of the article, and approved the final draft.
- Jia Qu conceived and designed the experiments, performed the experiments, analyzed the data, prepared figures and/or tables, authored or reviewed drafts of the article, and approved the final draft.

### Data Availability

Raw data is available at GitHub: https://github.com/CXLAZGitHub/NIRBMMDA.
Code and data are available in the Supplemental Files.

## Supplemental Information

Supplemental information for this article can be found online at http://dx.doi.org/10.7717/peerj.19637#supplemental-information.

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
