# Peer review of "Prediction of microbe-drug associations using a CNN-Bernoulli random forest model"

_PeerJ, doi:10.7717/peerj.19637_

## Round 0.1 · original submission · Major Revisions

· Academic Editor

Major Revisions

Three reviewers call for major revisions. They all agree that the selection of negative samples needs to be improved to avoid bias. Cross validation needs to be supplemented with an external case. The reviewers do agree that the paper describes what may be an important advance, but the complexity of the model calls for more details.

Reviewer 1 ·

Basic reporting

Language of the manuscript is good
Lack of lterature references in discussion section
Add more description in discussion section
Relevant prior literature should be appropriately referenced.

Experimental design

This manuscript presents a computational model (CNNBRFMDA) for predicting microbe-drug associations using a combination of a convolutional neural network (CNN) for feature extraction and a Bernoulli Random Forest (BRF) for association prediction. It demonstrates high predictive accuracy through 5-fold cross-validation and two case studies, outperforming other models. The integration of CNN and BRF enhances the predictive precision and interpretability, offering a cost-effective method to accelerate drug discovery and combat antibiotic resistance. However, the datasets used, though robust, are relatively small. Expanding these could further validate the model’s generalizability. While CNN is a powerful tool for feature extraction, the complexity of the CNN model might limit interpretability for some end-users. The random selection of negative samples could introduce bias; careful selection of negative samples may enhance model performance.

Validity of the findings

The combination of CNN for dimensionality reduction and BRF for prediction is innovative, enhancing model performance compared to existing methods.
The model demonstrates high AUC values of 0.9017 and 0.9146 for the MDAD and abiofilm datasets, respectively, indicating superior predictive capabilities.
The model retains performance with different datasets and case studies, confirming its reliability across various microbial and drug pairs.
This model provides a computational tool for identifying potential drug candidates, especially important in combating antibiotic resistance.
The manuscript includes thorough performance comparisons, case studies, and precision-recall analysis.

Additional comments

The following questions are intended to provoke further discussion, improvement, and potential validation of the model's utility in practical applications and incorporate description in the revised manuscript.
How would the performance of CNNBRFMDA change with larger, more diverse datasets or datasets from other biological contexts, such as viruses or fungi?
How can the interpretability of the CNN layer outputs be improved for researchers unfamiliar with deep learning models?
How does the model guard against overfitting, particularly given its complexity, and are there further steps (e.g., cross-validation techniques) that could be implemented to reduce this risk?
How does the random selection of negative samples (non-associated microbe-drug pairs) impact the model's overall performance and could more refined negative sampling methods be tested?
Can the computational cost of the CNN model be reduced further, perhaps by experimenting with different architectures or fewer parameters?
How well would the CNNBRFMDA model perform in a real-world setting with new, unseen microbes or drugs that are not present in the current datasets?
Why were Ciprofloxacin and Moxifloxacin chosen for the case studies, and how might the model's performance vary with other drug classes?
How can the findings from this computational model be translated into actual clinical or pharmaceutical trials, and what would be the practical steps for implementation?
What modifications could be made to the BRF method to further enhance its consistency or performance, particularly regarding handling complex associations?
How does the model address evolving microbial resistance mechanisms, and could the model be adapted to predict resistance trends rather than just drug-microbe associations?

Reviewer 2 ·

Basic reporting

No comment

Experimental design

No comment

Validity of the findings

No comment

Additional comments

The manuscript presents a computational model, CNNBRFMDA, for predicting microbe-drug associations. It combines Convolutional Neural Networks (CNN) with Bernoulli Random Forest (BRF) to improve feature extraction and prediction accuracy. The model is tested on two datasets, MDAD and abiofilm, demonstrating high predictive performance through 5-fold cross-validation and case studies involving ciprofloxacin and moxifloxacin. The manuscript offers a methodological advancement by integrating CNN and BRF but requires further clarification and enhancement in certain areas.
Overall Recommendation: Accept with Major and Minor Revisions
The manuscript presents a novel integration of CNN with BRF and offers strong empirical results, but it needs revisions to clarify its novelty, address methodological limitations, and improve the quality of writing and presentation.
Major Revision Points
1. The manuscript presents the integration of CNN and BRF as a novelty, but the specific advantages over existing models are not adequately discussed. Provide a stronger theoretical justification for the CNN-BRF combination. Clarify how this integration improves upon existing methods (e.g., traditional Random Forest or CNN-only models) in terms of accuracy, interpretability, or computational efficiency.
2. The manuscript does not address the generalization of the model to other biological domains beyond microbe-drug associations. Add a discussion on the generalizability and scalability of CNNBRFMDA to other types of biological associations (e.g., gene-disease, protein-ligand). Consider validating the model on an independent, unrelated dataset to demonstrate versatility.
3. The use of CNN introduces interpretability challenges, which are only partially addressed by the addition of BRF. Provide more details on how BRF improves the interpretability of the CNN model. Consider adding feature importance scores or visual aids such as heatmaps or attention maps to explain how the model derives its predictions.
4. The random selection of negative samples could introduce bias, and the manuscript lacks a robust discussion on handling data imbalance. Discuss strategies for mitigating data imbalance, such as weighted loss functions or oversampling methods like SMOTE. Clarify how negative samples were selected and the potential impact of selecting unverified associations as negatives.
5. The evaluation is primarily based on cross-validation, which may not fully represent real-world performance. Validate the model using an external test set to better assess generalizability. Include additional metrics (e.g., precision, recall) and real-world examples to demonstrate the practical utility of the model's predictions.
6. The case studies focus on predicting microbe associations for ciprofloxacin and moxifloxacin but lack critical analysis of the biological significance of the results. Expand the analysis of the case study results. For each predicted association, explain the biological relevance and validate the findings against experimental data or literature. Discuss mispredictions and potential areas for model improvement.
7. The manuscript claims computational efficiency but does not provide data to support these claims. Include a detailed comparison of the computational costs (e.g., memory usage, training time) of CNNBRFMDA versus other models. Demonstrate how the model scales with increasing data size.
8. The manuscript compares CNNBRFMDA with other advanced models but lacks baseline comparisons. Include comparisons with simpler baseline methods (e.g., logistic regression, k-nearest neighbors) to illustrate how much improvement CNNBRFMDA offers over traditional approaches.
9. The discussion of limitations and future directions is brief and lacks depth. Provide a more comprehensive discussion of limitations (e.g., reliance on known associations, potential for overfitting). Expand on future directions, such as the integration of additional data types or more sophisticated neural network architectures.
10. Representativeness of the datasets is not fully explored. Analyze the biases in the datasets used (e.g., geographic, species diversity) and their potential impact on model performance. Discuss how representative the MDAD and abiofilm datasets are for broader biological contexts.
Minor Revision Points
1. Consider simplifying the title to something more concise, such as "Prediction of Microbe-Drug Associations Using a CNN-Bernoulli Random Forest Model."
2. Include dataset names and performance metrics in the abstract for greater clarity on the evidence supporting the study.
3. Introduction: Streamline sentences and avoid redundancy. Improve integration of references by providing specific contributions of each citation.
4. Methods: Clarify the mathematical notations in feature vector construction. Provide details on how hyperparameters were tuned.
5. Summarize the performance comparison of models more concisely and avoid repetitive information. Provide specific examples of mispredictions and potential causes.
6. Expand on the potential real-world applications of the model and discuss the limitations of CNN models in bioinformatics.
7. Ensure consistency in the terms used (e.g., "biofilm dataset" vs. "abiofilm dataset"). Improve clarity by fixing minor grammatical issues.
8. Verify that all references follow the journal's citation format and correct any inconsistencies.
9. Improve the resolution of all figures.
10. The scientific name of an organism should be italicized.
Figure 1: The flowchart is clear but could benefit from more detailed labels for the input data types. Add labels to clarify data preparation steps and consider color-coding for clarity.
Figure 2: The figure could use more explanation for how features are split. Add annotations to explain random attribute bagging and feature splitting more clearly.
Figure 6 & 7: The precision-recall curves need more context for interpretation. Add reference lines or labels to clarify acceptable performance ranges.
Tables 1, 6-7:
The table headings need to be changed
The manuscript presents a strong methodological advancement in predicting microbe-drug associations through the integration of CNN and BRF. However, to improve clarity, scientific rigor, and real-world applicability, the suggested major and minor revisions should be addressed. With these improvements, the manuscript would make a valuable contribution to computational biology and bioinformatics.

Reviewer 3 ·

Basic reporting

The manuscript titled Microbe-Drug Association Prediction with Bernoulli Random Forest based on Convolutional Neural Network discusses the development of a novel computational model named Convolutional Neural Network with Bernoulli Random Forest for Microbe-Drug Association prediction (CNNBRFMDA), which has the features of both Convolutional Neural Network (CNN) and Bernoulli Random Forest (BRF), predicting association between microbe and drug to address microbial resistance challenges.

The model aims to construct feature vectors from known microbe-drug associations, microbe similarity, and drug similarity, using CNN for dimensionality reduction and BRF for prediction. The model has been evaluated on Microbe-Drug Association Database (MDAD) and abiofilm dataset, showing high mean AUC and std. deviation of 0.9017 +/- 0.0032 (MDAD) and 0.9146 +/- 0.0041 (abiofilm) respectively in five-fold cross-validation. Further, case studies on Ciprofloxacin and Moxifloxacin demonstrated the reliability of the model.
Overall, this study is a valuable contribution to the field of drug discovery and machine learning.

Experimental design

However, the following points are worth mentioning to enhance the scientific impact of the study:
1. Random selection of negative samples, i.e., microbe-drug pairs with no known association introduces significant potential biases in the dataset as there may be a number of yet undiscovered associations. How would you remove this bias and does this removal significantly impact the model accuracy?
2. What could be the different datasets for which this model is suitable? Have you tried this model on the expanded datasets? If yes, then what is the accuracy of the model?
3. In this study, the CNN model used 16 filters. Why?
4. What complexities need to be addressed while increasing the CNN filters?
5. The reason why CNNBRFMDA outperforms the other five models hasn’t been explained in detail, particularly based on different dataset characteristics.
6. Sensitivity analysis can be performed to identify potential weaknesses in the prediction of the model.

Validity of the findings

7. The discussion section doesn’t have any citations of related literature and doesn’t show a prior linkage to any kind of similar study or previously made models.
8. Future work and real-life applications can be discussed in detail.

Additional comments

9. There are a few grammatical errors and typos in the document.
10. A list of abbreviations needs to be made. Full form MDAD?
11. Improve the resolution of all figures.
12. Figure 1 should be labeled properly.
13. In Table 6 and Table 7, shouldn’t there be Microbe name column instead of Drug name column? Also, the table legend is quite confusing.

---

## Round 0.2 · accepted · Accept

· Academic Editor

Accept

In my view, all the issues pointed out by the reviewers were adequately addressed, and the manuscript was amended accordingly. Therefore, the revised version is acceptable now.